# Data augmentation guided Decouple Knowledge Distillation for low-resolution fine-grained image classification

## Abstract

Continuous development of convolutional neural networks has shown good performance for fine-grained image classification by identifying fine features in high-resolution images. However, in the real world, many images are due to camera or environmental restrictions. Low resolution images with fewer fine features result in a dramatic reduction in classification accuracy. In this study, a twophase Data Augmentation guided Decoupled Knowledge Distillation (DADKD) framework is proposed to improve classification accuracy for low-resolution images. In the proposed DADKD, one phase is data augmentation that generates a composite image and corresponding labels. Another stage is knowledge distillation, which minimizes differences between high-resolution and low-resolution image features. The proposed DADKD validated on three fine-grained datasets (i.e Stanford-Cars, FGVC-Aircraft, and CUB-200-2011 datasets). Experimental results show that our proposed DADKD achieves 88.19%, 78.98% and 80.33% classification accuracy on these three datasets, state-of-the-art methods such as SnapMix and Decoupled Knowledge Distillation (DKD). The method proposes a viable solution for fine-grained classification at low resolution.

## 1 Introduction

With the development of deep convolutional neural networks, impressive results have been achieved in computer vision tasks such as super-resolution image reconstruction(Wang et al., 2021; Kong et al., 2021), image classification(Krizhevsky et al., 2017; Simonyan & Zisserman, 2014), object detection(Li et al., 2020; Huang et al., 2022), tracking(Ren et al., 2015; Redmon et al., 2016), and image segmentation(Kim et al., 2022; Zhang et al., 2022). Fine-grained image classification is a challenging issue in classifying identical species into different subclasses, such as distinguishing between species of wild birds and vehicle models. Existing fine-grained classification models such as Inception(Szegedy et al., 2015) and ResNet(He et al., 2016) are typically trained on high-quality, high-resolution fine-grained datasets. However, in real-life applications, images collected from a distance may be blurred or low resolution. The performance of models trained on high resolution fine grained images cannot be guaranteed when applied directly to these low resolution fine grained images.

Low resolution fine grained image classification is quite challenging as it has minimal information content and few useful features can be extracted. For example, identifying details such as pedestrian faces or license plate numbers from an image is not easy because the associated feature extraction is difficult. There are three main approaches to classifying low resolution fine grained images: mixed resolution training approach(Wu et al., 2022), super resolution reconstruction approach(Dong et al., 2014), and knowledge distillation(Chen et al., 2022) based approach. The mixed-resolution training approach developed by Zangeneh et al.(Zangeneh et al., 2020) maps high-resolution and low-resolution images to a shared space using two deep convolutional neural networks. However, the hybrid resolution-based training approach only makes the model more friendly to multi-scale information perception, with limited improvement in recognition accuracy. In super-resolution reconstruction methods, GAN(Goodfellow et al., 2020) networks are often used to produce more realistic results. Wang et al.(Wang et al., 2018) used a simple classification network that was used as the discriminative model and a SRResNet(Ledig et al., 2017) backbone network as the generative model

to minimize the gap between reconstructed and authentic images. But the super-resolution approach is more expensive to train and maintain, and the accuracy of the recognition model depends on the output of the super-resolution model. There is no guarantee that the information output from the super-resolution model will be helpful for the classification task. The third approach, knowledge distillation, uses high-resolution images to train the teacher network and low-resolution images to train the student network(Zhu et al., 2019). Based on the generality of the knowledge distillation paradigm, this method is widely used in various deep learning models. Nevertheless, the improved recognition accuracy of this type of method is limited.

To improve the classification accuracy of low resolution fine grained images, a DADKD approach is proposed. Our approach involves two steps: Firstly, hybrid HR images generation, to label weights in HR images, channel focus mechanism and SnapMix data augmentation method was utilised; Predictive values of teacher and student models are minimised using a DKD method based on composite images while performing data augmentation knowledge transfer. Our main contributions are as follows:

- A novel data augmentation-based hybrid knowledge distillation framework is proposed for low-resolution fine-grained classification. This framework utilizes hybrid images and label information to guide the difference between teacher and student models. Improving recognition accuracy of fine-grained models at low resolution.

- This study developed data augmentation method that combines the channel attention mechanism with Snap-Mix, along with a DKD to generate hybrid images, which effectively enhances the accuracy of classification recognition for low-resolution fine-grained images.

- The proposed DADKD achieved 88.19%, 78.98% and 80.33% classification accuracy respectively on Stanford-Cars, FGVC-Aircraft, and CUB-200-2011 datasets, which significantly outperformed the benchmark models.

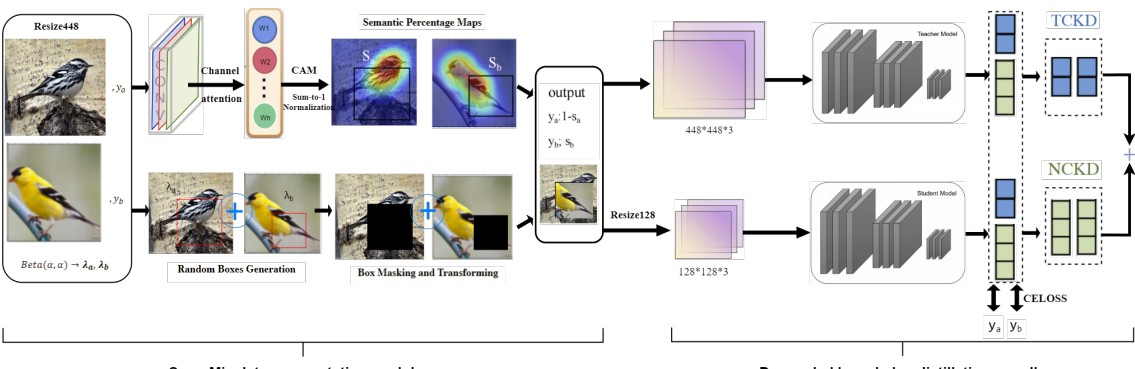

Figure 1: The proposed data augmentation-guided knowledge distillation for low-resolution fine-grained classification.

## 2 RELATED WORK

### 2.1 DATA AUGMENTATION

In general, the better the quality and quantity of data used for training, the better the generalization capability of the model. The most common offline augmentation methods are cropping, flipping, rotating, scaling, shifting, Gaussian noise, color fading, etc. Online augmentation is more suitable for large data sets , which typically merge two images and generate corresponding labels based on their merging method. For example, Mixup(Zhang et al., 2017) is a pixel-based augmentation method in which two random samples are proportionally mixed, and the classification results are proportionally distributed. Cutout(DeVries & Taylor, 2017), on the other hand, cuts out any part of the sample and fills it with 0 pixel values. CutMix(Yun et al., 2019) does not fill the cutout region with 0 pixels but randomly samples pixel values from other parts of the training set to fill the cutout

region. The classification result is proportionally distributed according to the split of the two images, which are two data augmentation methods based on segmentation. This method can force the model to classify from a local (subtle) perspective, thus increasing the sensitivity of the model to subtle features. Inspired by this, data augmentation can be used in fine-grained image classification tasks.

## 2.2 KNOWLEDGE DISTILLATION

Knowledge distillation offers new ideas for deploying high-quality models in limited storage space, effectively transferring teacher knowledge to student models. Hinton et al.(Hinton et al., 2015) introduced the concept of knowledge distillation, which involves instructing a smaller network of students using a larger network of teachers. Park et al.(Park et al., 2019) mentioned that attention should be paid to the structure between categories rather than the categories themselves during distillation. Zhao et al.(Zhao et al., 2022) reconstructed the logit-based distillation method and Yuan et al.(Yuan et al., 2020) revealed the relationship between knowledge distillation and label smoothing regularization. Through knowledge distillation, the student model can obtain the discriminative information of the teacher model for subtle feature areas,to improve the discrimination ability of blurred areas for low resolution fine grained images.

## 3 PROPOSED METHOD

To improve the classification accuracy of low resolution fine grained images, the DADKD framework was proposed in this study. As depicted in Figure 1, the proposed DADKD comprises two modules: the Snap-Mix data augmentation module and the decoupled knowledge distillation module. The data augmentation module uses a channel attention mechanism to enhance the model's capacity to localize subtle features through semantic data blending, prompting the model to recognize targets based on local (subtle) features. The knowledge distillation module, following a teacher-student paradigm, extracts visual features from blended images. Here, the student model learns from the teacher model on how to discern fine-grained features in high-resolution images, thereby enhancing its ability to distinguish fine-grained features in the blurred regions of low-resolution images.

More specific, for the given images $I_i \in R^{3 \times W \times H}$ from original dataset $\{(I_i, y_i) \mid i \in [0, 1, ..., N-1]\}$ were resized to a resolution of $448 \times 448$ and feed it into the data augmentation module, where a synthetic image $\tilde{I}$ is generated by irregularly blending a pair of images and generating the label $y_i$ weights $\rho_a$ and $\rho_b$ corresponding to the synthetic image based on its attention map.Here, $\rho_a$ and $\rho_b$ correspond to the label $y_a$ and $y_b$ respectively. The composite image is then resized to $128 \times 128$ resolution, fed into the student model, and fed the original synthetic images into the teacher model. We leverage the label weights of the original synthetic image to steer the hard loss of the student model, and the decoupled classification results of the teacher model to guide the soft loss of the student model. This approach enhances the accuracy of the student recognition model. The framework distinguishes itself from existing approaches in three main ways: 1) Incorporating a channel attention mechanism in class activation mapping to replace feature map weights generated by global averaging pooling in SnapMix; 2) Guide the hard loss of the student model through label weights generated by the data augmentation module. It can also be described as a transfer of data augmentation knowledge ; 3) Improve the DKD method to better accommodate blended images.

## 3.1 DATA AUGMENTATION MODEL

SnapMix uses the class activation map generated from the original image, which allows asymmetric blending operations to ensure semantic correspondence between the synthetic image and the blended labels Huang et al. (2021). In our study, SnapMix is employed for fine-grained semantic scale blending, it blend an image by cropping a region from one image and transforming and pasting it into another image at a random location. The blending operation is shown as follows.

$$\tilde{I} = (1 - M_{\lambda^a}) \odot I_a + T_\theta (M_{\lambda^b} \odot I_b) \tag{1}$$

Let $M_{\lambda^a}$ and $M_{\lambda^b}$ are binary masks with random box regions with area ratios $\lambda_a$ and $\lambda_b$, and $T_\theta$ transforms $I_b$'s cutout region to match $I_a$'s box region.

On the other hand, as fine grained image recognition often involves subtle differences between targets, generating a class activation map of the image often go a long way to avoid introducing significant noise (Huang et al., 2021). To calculate the semantic composition of the input image, the SnapMix method generates feature map weights using global average pooling (GAP) to normalize the image's class activation map (CAM) to produce a semantic percentage map (SPM). Then the weights of the two original images in the hybrid image are obtained based on SPM and $\lambda$. In contrast to the SnapMix approach, the channel focus mechanism is added to the data augmentation module to calculate the weights of the feature maps, using a fully connected neural network (two fully connected layers, ReLu and sigmoid) to perform a non-linear transformation of the post-GAP results to obtain more accurate classifier weights. Finally, the semantic composition of the hybrid image is estimated based on the semantic relevance of each original image pixel to its corresponding label.

## 3.2 KNOWLEDGE DISTILLATION MODEL.

Knowledge distillation is a concept of dark knowledge extraction that can be understood in terms of transfer learning and model compression(Hinton et al., 2015). The focus is on proposing a soft target loss to complement the hard target loss. Soft target loss is the loss calculated by the KL function between the predicted result of the student model and the predicted result of the teacher model. On the other hand, the hard target loss is the loss calculated between the predicted result of the student model and the original one-hot label via the Cross-entropy loss function. The loss function of the knowledge distillation paradigm is expressed as

$$L = aL^{(soft)} + (1 - a) L^{(hard)} \tag{2}$$

where a is a hyperparameter between 0 and 1, $L^{(soft)}$ stands for soft target loss and $L^{(hard)}$ stands for hard target loss.

As shown in the knowledge distillation module in Figure 1, we resize the hybrid image generated in the data augmentation module to a resolution of $128 \times 128$, send it to the student model, send the hybrid image with a resolution of $448 \times 448$ to the teacher model, and use the two label weights generated by the data augmentation module with the prediction results of the student model to calculate the hard target loss, and then the teacher model with the predictions of the student model to calculate the soft target loss. These two components are described in detail below.

**Loss with hard target**: We define $log^T$ as the logit generated by the hybrid HR image from the teacher model and $log^S$ as the logit generated by the hybrid LR image from the student model. $(\Theta)$ is the cross-entropy loss function, and $\rho_a$ $\rho_b$ is the weight of the two labels generated by the HR image through the data augmentation module. The hard loss of the DADKD model is defined as

$$L^{(hard)} = \rho_a \times \Theta \left( log^S, y_a \right) + \rho_b \times \Theta \left( log^S, y_b \right) \tag{3}$$

This paper applies label weights generated from HR image data augmentation to LR images, allowing the student model to better accept the teacher model's predictions of HR blended image weights, which we define as data augmentation knowledge transfer.

**Loss with soft target**:The soft target loss within the knowledge distillation paradigm is typically referred to as KD. The DKD approach seeks to analyze KD more comprehensively by decoupling model predictions into target and non-target classes. Target classes are the categories we anticipate the model to identify, while non-target classes are the converse. Distillation for the target class and the non-target class is termed Target Class Knowledge Distillation (TCKD) and Non-target Category Knowledge Distillation (NCKD), respectively. This allows us to rederive the loss formula for KD to acquire new equivalent expressions.

$$KD = TCKD + \left( 1 - p_t^T \right) NCKD \tag{4}$$

When the new expression for KD is reviewed, it is found that the loss corresponding to NCKD is coupled with a weight $1 - p_t^T$, $p_t^T$ is the teacher's level of confidence in the target class. In other words, the more confident the teacher network is in its predictions, the less weight NCKD will be and the less influence it will have. Neverless, the relationship between non-target classes is crucial to the knowledge distillation framework; NCKD may be the main reason for KD's effectiveness(Zhao et al., 2022). By decoupling the traditional knowledge distillation, a new knowledge distillation paradigm is generated to increase the weight of NCKD in the overall knowledge distillation. i.e., DKD, with the following expression:

$$DKD = \alpha TCKD + \beta NCKD \tag{5}$$

where $\alpha$ and $\beta$ are hyperparameters, In the experiment it was set to 1:8. The decoupled knowledge distillation approach decouples the predictions of the teacher model from those of the student model and then distils them separately.

After the data augmentation module, a mask is applied to extract these two target classes, which is defined as $TCKD_{mix}$. Assuming that the class with the classification task is $N$, the class corresponding to $TCKD_{mix}$ is 2, and the class corresponding to NCKD is $N - 2$. The soft loss of the knowledge distillation module of the DADKD framework is defined as

$$L^{(soft)} = \alpha TCKD_{mix} \left(log^T, log^S\right) + \beta NCKD \left(log^T, log^S\right) \tag{6}$$

## 4 EXPERIMENT SETUP

Our experiments were conducted with three standard fine-grained datasets: CUB-200-2011(Wah et al., 2011), Stanford-Cars(Krause et al., 2013), and FGVC-Aircraft(Maji et al., 2013). To simplify notation, use the short names CUB, Cars, and Aircraft throughout the rest of the paper. Multiple network architectures (Resnet 18, 34, 50, 101) were used as baselines to evaluate our approach. Based on each network architecture, our method's performance compare with related knowledge distillation and data augmentation methods. Our results also compare with current state-of-the-art methods for fine-grained image identification.

**Baselines and backbone networks.** Four network backbones were used as a baseline against which to compare our approach with other methods. Whenever not otherwise specified, we refer to the baseline as a pre-trained neural network model that has been fine-tuned on the target dataset based on the Imagenet dataset. This study used Resnet 18, 34, 50, and 101 network structures, and these experiments were adapted from the TorchVision package.

**Data augmentation and knowledge distillation methods.** This study compared our approach with two representative knowledge distillation methods and one data augmentation method: KD, DKD, and SnapMix.These methods were implemented based on published code and experimented on low-resolution fine-grained datasets, as previous work did not have classification results for low-resolution datasets.

**Implementation details.** Pytorch deep learning framework was used to implement all deep learning models in this study. An Intel Xeon Platinum 8160T@2.1 GHz was used for the experiments on an Ubuntu 18.04.6 server. A NVIDIA RTX A5000 GPU accelerated it with 24 GB of RAM.

**Training details.** The train weights were learned using stochastic gradient descent (SGD) with a momentum of 0.9, and the novel parameters were learned with a base learning rate of 0.001. Every 80 epochs, decayed the learning rate by 0.1 and trained our model for 200 epochs.

## 5 EXPERIMENTAL RESULTS AND ANALYSIS

### 5.1 DIFFERENT FINE-GRAINED IMAGE CLASSIFICATION MODEL COMPARISON

This section compares the performance of DADKD with other state-of-the-art fine-grained recognition techniques. As shown in Table 1, we can find that the recognition accuracy of the Resnet network improves with deeper network layers, and Resnet101 outperforms Resnet50 on all three

Table 1: The accuracy (%) comparison with state-of-the-art methods on CUB, Cars, and Aircraft. For the baselines and our approach, we reported their average accuracy of the final ten epochs and showed their best accuracy in the brackets.

| Method | image Size | Accuracy(%) | | |
| --- | --- | --- | --- | --- |
| | | CUB | Cars | Aircraft |
| Mcloss(VGG) | 128 | 79.19 | 84.56 | 80.41 |
| Mcloss(Res50) | 128 | 74.81 | 76.48 | **82.31** |
| PCA-net | 128 | 65.30 | 67.96 | 70.62 |
| PMG | 128 | 60.21 | 64.53 | 62.13 |
| Inceptionv3(299) | 299 | 66.12 | 81.97 | 72.12 |
| API-net | 128 | 68.67 | 74.81 | 63.91 |
| MG | 128 | 63.55 | 70.01 | 67.39 |
| Res50 | 128 | 66.40(66.76) | 69.25(69.34) | 58.44(58.62) |
| Res101 | 128 | 67.67(67.88) | 72.35(72.51) | 64.91(65.11) |
| Rese50+DADKD | 128 | 78.22(78.31) | 87.74(87.90) | 72.63(72.87) |
| Res101+DADKD | 128 | **80.33(80.46)** | **88.19(88.31)** | 78.98(79.11) |

fine-grained datasets, with the aircraft dataset showing the most significant performance improvement. Based on this result, A deeper model can better handle label noise.

The improved baseline using DADKD is comparable to some of the fine-grained image recognition methods that require the most complex design and lengthy inference. To capture standard discriminative features, PCA-Net (Zhang et al., 2021) encourages interactions between feature channels of pairs of images in the same class to compute channel similarity. API-Net (Zhuang et al., 2020) is forced to focus on other discriminative regions after removing salient regions enhanced by channel interactions. In high-resolution fine-grained image recognition, these complex networks can achieve high recognition accuracy. Neverless, as the image resolution decreases and the delicate features become blurred, the complex networks cause the model recognition accuracy to drop dramatically. As shown in Table 1, the recognition accuracy of complex networks is even worse than that of baseline networks such as Resnet when facing low-resolution images.

In contrast, simpler designs such as Mcloss (Chang et al., 2021) and the DADKD framework achieve good results in low-resolution fine-grained recognition, probably because simple network structures are suitable for image with little vision information. Combined with the Resnet-101 backbone network, DADKD achieves recognition accuracies of 80.33%, 88.19%, and 78.98% for CUB, Cars, and Aircraft, respectively, in the testing phase without any additional features, outperforming most existing techniques. Our method achieves the highest recognition accuracy on the CUB and car datasets and approaches the state-of-the-art Mcloss method (Resnet50) on the aircraft dataset. In addition, the Inceptionv3(Krause et al., 2016) network inputs images at a minimum resolution of $299 times 299$, which is also used in this experiment.

Table 2: Performance comparison (Mean Acc.%) of methods using backbone networks Resnet-18 and Resnet-34 on fine-grained datasets. Each method's improvement over the baseline is shown in the brackets.

| | CUB | | Aircraft | | Cars | |
| --- | --- | --- | --- | --- | --- | --- |
| | Res18 | Res34 | Res18 | Res34 | Res18 | Res34 |
| Baseline | 59.44 | 62.79 | 50.22 | 52.41 | 60.58 | 64.73 |
| SnapMix | 65.41(+5.97) | 66.63(+3.84) | 59.71(+9.49) | 61.32(+8.91) | 71.68(+11.1) | 77.45(+12.72) |
| KD | 62.81(+3.37) | 65.52(+2.73) | 53.67(+3.45) | 55.89(+3.48) | 61.02(+0.44) | 65.12(+0.39) |
| DKD | 71.31(+11.87) | 67.83(+5.04) | 58.63(+8.41) | 59.67(+7.26) | 75.97(+15.39) | 79.63(+14.9) |
| DADKD | **73.31(+13.87)** | **69.90(+7.11)** | **65.61(+15.39)** | **68.52(+16.11)** | **80.61(+20.03)** | **85.34(+20.61)** |

Table 3: Performance comparison (Mean Acc.%) of methods using backbone networks Resnet-50 and Resnet-101 on fine-grained datasets. Each method's improvement over the baseline is shown in the brackets.

| | CUB | | Aircraft | | Cars | |
|---|---|---|---|---|---|---|
| | Res50 | Res101 | Res50 | Res101 | Res50 | Res101 |
| Baseline | 66.40 | 67.67 | 58.44 | 64.91 | 69.25 | 72.35 |
| SnapMix | 71.32(+4.92) | 73.04(+5.34) | 63.87(+5.43) | 69.29(+4.38) | 81,54(+12.29) | 83.16(+10.08) |
| KD | 71.14(+4.47) | 72.71(+5.04) | 58.22(-0.22) | 64.68(-0.31) | 72.33(+3.08) | 75.42(+3.07) |
| DKD | 75.92(+9.52) | 76.38(+8.71) | 62.13(+3.69) | 68.01(+3.1) | 84.94(+15.69) | 85.69(+13.34) |
| DADKD | **78.22(+11.82)** | **80.33(+12.66)** | **72.63(+14.19)** | **78.98(+14.07)** | **87.74(+18.49)** | **88.19(+15.84)** |

## 5.2 COMPARISON WITH DATA AUGMENTATION AND KNOWLEDGE DISTILLATION

This section compares the performance of the DADKD, KD, DKD, and SnapMix methods on three fine-grained datasets: CUB, Aircraft, and Cars. The results are shown in Table 2-3. First, we can observe that our proposed DADKD method consistently outperforms similar methods. When using Resnet101 as the baseline model, it achieves the highest accuracy of 80.33%, 88.19%, and 78.98% on the CUB, Cars, and Aircraft datasets, respectively. It shows that it becomes more sensitive to fine-grained features as the depth of the model increases.

Comparing the recognition rate improvement produced by DADKD with different baseline models, it can be seen that smaller networks perform better, with Resnet18 achieving the highest recognition rate improvement of 13.78% in the CUB dataset, and Resne34 achieving the highest recognition rate improvement of 20.61% in the Cars dataset, which suggests that our framework is more suitable for combining with smaller models for applications.

It is worth noting that KD performs very poorly in the Aircraft dataset and even shows a decrease in recognition accuracy on the Resnet50 and Resnet101 baseline models, which we speculate may be due to model overfitting caused by the high number of model layers.

Meanwhile, comparing the three datasets revealed that all methods showed limited improvement on the aircraft dataset with the lowest recognition accuracy. It may be due to the fact that the aircraft dataset shows more subtle category differences, which increases the risk of noisy labeling and creates difficulties for model classification.

## 5.3 INFLUENCE OF HYPERPARAMETERS.

The hyperparameter $\alpha$ of the data augmentation module in the DADKD framework determines the size of randomly generated patches. Our experiments found that the accuracy rate fluctuated slightly as $\alpha$ only increased. In our experiments, $\alpha$ was set to 5 to ensure that the images were mixed with medium-sized boxes. Experimented with different distillation temperatures and soft with hard loss ratios in the knowledge distillation model. We performed a comparison test on the CUB dataset using Resnet18 as the baseline model, and the results are shown in Table 4.

Table 4: Comparison of different hyperparameters in the experiment.

| temp | 2:8 | 3:7 | 4:6 | 5:5 |
|---|---|---|---|---|
| 3 | 72.8% | 73.3% | 71.3% | 68.2% |
| 5 | 72.9% | 73.1% | 71.6% | 67.9% |
| 7 | 72.6% | 72.8% | 71.2% | 67.8% |

## 5.4 ABLATION STUDY

According to Table 4, the distillation temperatures were set at 3, 5, and 7. Identification accuracy fluctuates between 0.3 percent and 0.5 percent with increasing distillation temperature, Which suggests that the DADKD framework is insensitive to distillation temperature. At the same time, the

Table 5: Ablation experiments of DADKD.

| baseline(Res18) | Date augmentation model | Knowledge distillation model | Accuracy(%) |
|:---:|:---:|:---:|:---:|
| ✓ | | | 59.44 |
| ✓ | ✓ | | 68.41 |
| ✓ | | ✓ | 71.28 |
| ✓ | ✓ | ✓ | 73.31 |

soft and hard loss ratios have a more significant impact on the model. The model accuracy gradually decreases as the soft loss ratio continues to increase. At a distillation temperature of 3, the model identification accuracy was only 68.2% for the 5:5 ratio, indicating that hard losses dominate the model classification process and soft loss is a kind of complement to hard loss. Finally, the model achieves peak classification accuracy at a distillation temperature of 3 and a soft-to-hard loss ratio of 3:7.

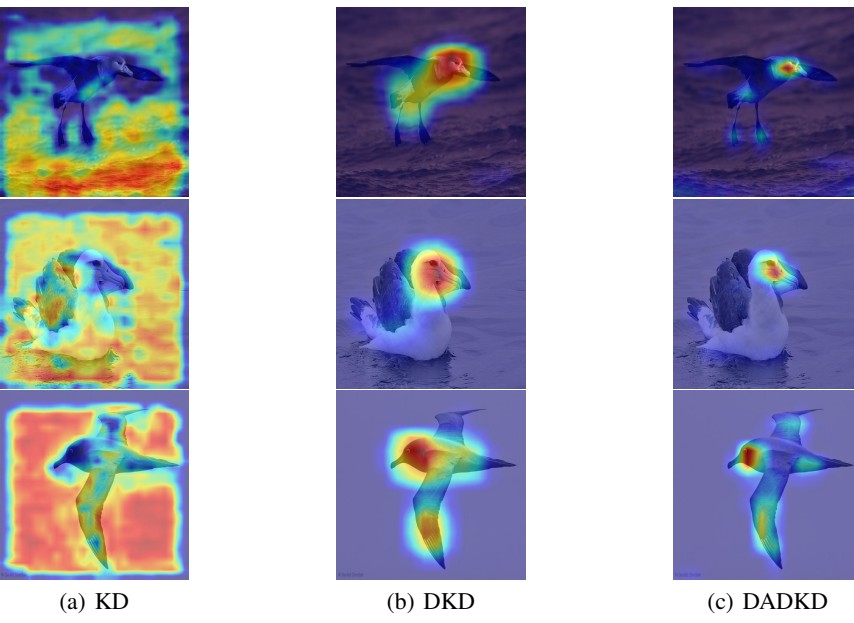

(a) KD       (b) DKD       (c) DADKD

Figure 2: CAM visualization of different Knowledge distillation methods.

The ablation experiments were done on the cub dataset, and we selected Resnet18 as the baseline network to compare the different modules in the framework; the results of the experiments are shown in Table 5 Compared to the baseline model(59.44%), adding the data augmentation module resulted in an accuracy of 68.41%, and adding the knowledge distillation module resulted in an accuracy of 71.28%, which shows that all the modules of this work positively affect the network and the knowledge distillation module dominates the framework compared to the data augmentation module.

## 5.5 VISUALIZATION

Figure 2 shows examples of CAMs correctly predicted by DADKD but incorrectly classified by KD and DKD. We can observe that some background patterns distracted the attention of KD, which may have contributed to the incorrect prediction. Meanwhile, compared with DKD, DADKD can

effectively enhance the baseline model sensitivity to the delicate features of low-resolution targets; that is, more attention is paid to the wings and beaks of birds.

# 6 CONCLUSION

In low resolution fine grained image recognition, it is essential to discern subtle features (e.g. bird wings, beaks) between low resolution image classes. To accurately recognize low-resolution fine-grained images by baseline models, this paper proposes a data augmentation guided decoupling knowledge distillation framework to improve recognition accuracy. Integrate the attention mechanism and SnapMix into the data augmentation model to improve the sensitivity of the baseline model to fine-grained features. In the knowledge distillation module, the improved decoupled knowledge distillation method is combined with the label weights generated by the data augmentation module to improve the baseline model's ability to discriminate fine-grained features in blurred regions of low-resolution images. Experiments on three fine-grained datasets, Cars, Aircraft, and CUB, show that the proposed framework achieves classification accuracy of 88.19%, 78.98%, and 80.33%, respectively, surpassing advanced methods such as SnapMix and DKD.

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
