# OpenReview forum: "Data augmentation guided Decouple Knowledge Distillation for low-resolution fine-grained image classification"
_ICLR.cc/2024/Conference — Submitted to ICLR 2024_

### Official Review · Reviewer_9NKv · 2023-10-18

**Soundness:** 2 fair
**Presentation:** 3 good
**Contribution:** 2 fair
**Rating:** 3
**Confidence:** 5

**Summary:**

This paper proposes a learning based fine-grained visual categorization method. The key idea is to first generate both low-resolution and high-resolution images for feature extraction, and then implement knowledge distillation between low- and high- resolution representation.
Experiments conducted on three standard FGVC benchmarks show the effectiveness of the proposed method.

**Strengths:**

- Overall this paper is well-written and easy-to-follow.

- The idea and technical design is straight-forward.

- The experiments indeed validate the proposed method's effectiveness, with a significant performance improvement.

**Weaknesses:**

- The technical novelties of this framework are somewhat incremental. Overall, the teacher and student network take the high- and low- resolution features as input and then knowledge distillation loss is implemented, which is not uncommon for knowledge distillation based framework, or some prior low-resolution classification works as the authors listed in the paper.

- About the loss weight in Eq.5. Indeed Table 4 gives an ablation study, but the ratio range is still too narrow, from the reviewer's view. What if the parameter $\alpha$ is larger, or the ratio is larger or smaller?
Besides, how is $\alpha$ in Eq.2 and $\rho$ in Eq.3 computed or determined? Besides, how does it impact the performance?

- The related work of this paper is limited and lack extensive discussion with prior works. As far as the reviewer concerns, some critical flaws remain:

(1) Some low-resolution FGVC methods and key references in the past few years are both missing and lack comparison, for example

[a] Cai, Dingding, et al. "Convolutional low-resolution fine-grained classification." Pattern Recognition Letters 119 (2019): 166-171.

[b] Yan, Tiantian, et al. "Discriminative feature mining and enhancement network for low-resolution fine-grained image recognition." IEEE Transactions on Circuits and Systems for Video Technology 32.8 (2022): 5319-5330.

[c] Yan, Tiantian, et al. "Discriminative information restoration and extraction for weakly supervised low-resolution fine-grained image recognition." Pattern Recognition 127 (2022): 108629.

(2) The related work subsection should include another subsection for FGVC and low-resolution FGVC to provide extensive discussion on this field.

(3) The FGVC works in FGVC community are nearly all missing.

- The state-of-the-art method comparison is not either extensive or favorable. Reasons:

(1) The above related low-resolution FGVC methods are missing, which should be compared and discussed extensively to show its effectiveness.

(2) The already compared FGVC methods in Table1 are also not extensive. PMG and API-Net are both before 2021. Some more recent FGVC methods in 2022 and 2023 should be compared.

**Questions:**

Q1: innovation and insights of the proposed framework. Is it significant enough compared with some prior knowledge distillation based low-resolution classification works?

Q2: More extensive ablation studies based on Eq.5.

Q3: Some implementation details on Eq.2 and 3 should be further clarified. Besides, if necessary, more ablation studies should be given.

Q4: Lack some prior low-resolution FGVC methods for both comparison and related work discussion.

Q5: The related work section is not well written or extensive.

Q6: FGVC method in the past few-years are all missing for either related work or comparison.

Q7: The compared methods are not very extensive or convincing.

---

### Official Review · Reviewer_c2Tf · 2023-10-24

**Soundness:** 3 good
**Presentation:** 3 good
**Contribution:** 3 good
**Rating:** 5
**Confidence:** 3

**Summary:**

This paper proposes a data augmentation-based hybrid knowledge distillation framework. The first stage is generating a composite image and corresponding labels. The second stage is knowledge distillation, which minimizes differences between high-resolution and low-resolution image features.

**Strengths:**

The target issues of the paper are meaningful and worth exploring.
This submission gives a valuable implementation of such an idea.

**Weaknesses:**

1. This paper is not clearly written. The content is not sufficient. If the main text of the submission achieves an upper limit of 9 pages, the paper can provide more experiments or descriptions of their methods.

2. The motivation is not clear. Why the author proposed these two modules has not been fully explained in the introduction section.

3. More experiments of resolution selection should be conducted, such as 64, 32.

4. (Zhu et al., 2019) have also utilized knowledge distillation to transfer knowledge from a high-resolution images trained teacher to a low-resolution student. A comparison is better.

**Questions:**

See Weaknesses.

---

### Official Review · Reviewer_EVNV · 2023-10-31

**Soundness:** 2 fair
**Presentation:** 3 good
**Contribution:** 2 fair
**Rating:** 3
**Confidence:** 4

**Summary:**

The paper presents a method called Data Augmentation guided Decoupled Knowledge Distillation (DADKD) for enhancing FGVC accuracy by using data augmentation that generates a composite image and corresponding labels. The knowledge distillation minimizes differences between high-resolution and low-resolution image features. The proposed approach uses two independent models (backbone and auxiliary) in decision-making. The approach is evaluated on three (CUB-200, Aircraft, and Stanford Cars) FGVC datasets and the performance is comparable to the state-of-the-art (SOTA). The method is suitable for FGVC at low resolution.

**Strengths:**

The idea is good and is inspired by the drawbacks of deep networks in capturing discriminative features to enhance the FGVC accuracy for low-resolution images.

The paper has justified the usefulness of data augmentation-guided knowledge distillation and its role in discriminating fine-grained categories.

Experimental evaluation using well-known benchmarked datasets of CUB-200, Standford Cars, and Aircraft datasets is carried out. On each dataset, the performance of the proposed approach is compared to the baselines and some recent approaches.

Ablation study involving the benefit of data augmentation and Knowledge distillation model via experimentation on the CUB-200 dataset.
The article also provides an interesting interpretation of the model via visualization using Grad-CAM.

**Weaknesses:**

The proposed idea is an integration of existing techniques to enhance FGVC accuracy for low-resolution images (128x128). However, the paper has some drawbacks (e.g., novelty, experimental comparison to recent works) and need to be addressed in order to justify the proposed approach in comparison to the SOTA approaches and advance knowledge in this direction. These drawbacks are:

The main drawback is the novelty. There are two components and are standard ones except the semantic percentage maps (SPM) and the loss TCKD/NCKD and thus, focus should have been on SPM, TCKD and NCKD to critically analyze model performance.

The paper claims that the proposed approach achieves SotA performance on three (CUB, Cars and Aircraft) datasets and is over-stated. The paper has missed a significant number of recent works and are:

CUB-200 dataset: SR-GNN (Bera et al., IEEE TIP 2022) 91.9%, CAP (Behera et al, AAAI 2021) 91.8%, Stacked LSTM (Ge et al., CVPR 2019) 90.4%, BARM 89.5% (Liu et al., IEEE Trans on Multimedia 2019)

Cars dataset: SR-GNN (Bera et al., IEEE TIP 2022) 96.1%, CAP (Behera et al, AAAI 2021) 95.7%, AutoAug (Cubuk et al., CVPR 2019) 94.8%, GPipe (Huang et al., NeurIPS 2019) 94.6% and DCL (Chen et al., CVPR 2019) 94.5%

Aircraft dataset: SR-GNN (Bera et al., IEEE TIP 2022) 95.4%,  CAP (Behera et al., AAAI 2021) 94.9%, Hierarchical Multi-Scale (Wharton et al., BMVC 2021) 94.9%, Graph-based high-order (Zhao et al., CVPR 2021) 94.3%, category-specific semantic coherency (Wang et al., ACM Multimedia 2020) 93.8%.

It would be nice to have a section on model capacity and computational complexity (e.g. Params, GFLOPS, per-image inference time, etc) to further improve the article. This should be compared to the other SOTA models.

Many approaches use the input resolution 224x224 (e.g., ResNet-50 and MobileNetV2) and achieve SOTA accuracy. The proposed approach uses double this size (448x448) for Teacher and 128x128 for students. Is 128X128 considered low-resolution? It would be interesting to see the performances for 64x64 resolution.

The paper is missing an important comparison to recent SOTA approaches (CAP: context-aware attentional pooling, Behera et al., AAAI 2021; SR-GNN, Bera et al., IEEE TIP 2022) for ResNet-50 backbone and achieves significantly higher accuracy for the input size of 224x224 with single stage end-to-end training. CAP achieved 89.2 (CUB), 94.0 (Cars), and 94.4 (Air) using MobileNet V2 as a backbone. Justification and comparative discussions could significantly improve the quality of the paper.

Contribution from $\rho_a$, $\rho_b$, TCKD, and NCKD should be evaluated and justified.

“The performance of models trained on high-resolution fine-grained images cannot be guaranteed when applied directly to these low-resolution fine-grained images.” Need proof and justification.

“Our results also compare with current state-of-the-art methods for fine-grained image identification.” The SOTA approaches achieved significantly above the results achieved by the proposed approach. The authors have missed the SOTA approaches.

**Questions:**

A clear description of novelty and advancement of knowledge is required.

Comparisons to the SOTA approaches need to be justified and consider more recent approaches for FGVC as outlined in the "Weaknesses" section. Many important references are missing.

The role of TCKD, NCKD, $\rho_a$, and $\rho_b$ should be experimentally validated and justified.

Many SOTA approaches use input size 224x224. The use of 128x128 size will be considered as a low-resolution? The SOTA performances using 224x224 size is significantly higher (> 90%) than the achieved results. Any justification/explanations?

The model capacity and computational complexity (e.g. Params, GFLOPS, per-image inference time, etc)  in comparison to SOTA?

---

### Official Review · Reviewer_XNWE · 2023-11-01

**Soundness:** 2 fair
**Presentation:** 1 poor
**Contribution:** 1 poor
**Rating:** 3
**Confidence:** 3

**Summary:**

This paper aims at low-resolution image classification and introduces a two-phase framework, which first utilizes SnapMix (S Huang et al. 2020) for data augmentation and then knowledge distillation for improving performance. Specifically, in the knowledge distillation stage, the augmented high-resolution images are fed into a teacher network while the corresponding low-resolution images are for a student network. The student network learns to predict categorical probabilities that are similar to those from the teacher network, as in previous knowledge distillation strategies. Experiments on three datasets show that the proposed method improves the performance of student networks.

**Strengths:**

+ The experimental results show that the proposed method does bring performance gains.

**Weaknesses:**

- Lack of novelty. The proposed method adds an existing attention module in SnapMix and two hyperparameters in the loss term of knowledge distillation, yet without providing reasonable and insightful motivations. The proposed method is called "Data augmentation guided Decouple Knowledge Distillation", however, I cannot understand where and how the guidance works, as the proposed method uses these two methods separately. As to using different loss weights, extensive methods have been proposed in related research, e.g., hard example mining and focal loss.

- The experiments are insufficient. State-of-the-art methods on low-resolution image recognition should be included for comparison. Besides, the performance of the teacher network should be reported as well.

- This paper must be polished carefully, as it has many typos and grammatical errors. To name a few, "it blend", "go a " (should be "goes"), and "Resne34" (Resnet34). This makes the paper hard to follow, particularly in Sec. 3.2.

**Questions:**

Please refer to the weakness section for the details.

---

### Meta-Review · Area_Chair_NY9e · 2023-12-11

**Metareview:**

This paper proposes a two-stage approach to low resolution fine-grained image classification.  The first stage is data augmentation (SnapMix) and the second stage is knowledge distillation via teacher-student learning.

The strengths of the paper are the topic of wide interests and demonstrated performance gains.  The weaknesses of the paper are lack of technical novelty, insufficient experimental validation, and poor writing.

The paper has received 4 reviews, with ratings of 3/3/5/3.  There are no rebuttals.

The AC recommends reject.

**Justification For Why Not Higher Score:**

Consensus of rejection.
Lack of novelty.
Insufficient experimental validation.

**Justification For Why Not Lower Score:**

N/A

---

### Decision · Program_Chairs · 2024-01-16

Reject